# IL-35 and RANKL Synergistically Induce Osteoclastogenesis in RAW264 Mouse Monocytic Cells

**DOI:** 10.3390/ijms21062069

**Published:** 2020-03-18

**Authors:** Yosuke Kamiya, Takeshi Kikuchi, Hisashi Goto, Iichiro Okabe, Yuhei Takayanagi, Yuki Suzuki, Noritaka Sawada, Teppei Okabe, Yuki Suzuki, Shun Kondo, Jun-ichiro Hayashi, Akio Mitani

**Affiliations:** Department of Periodontology, School of Dentistry, Aichi Gakuin University, 2-11 Suemori-dori, Chikusa-ku, Nagoya 464-8651, Japan; k-yosuke@dpc.agu.ac.jp (Y.K.); hisashi@dpc.agu.ac.jp (H.G.); iichiro@dpc.agu.ac.jp (I.O.); ag173d14@dpc.agu.ac.jp (Y.T.); yukip@dpc.agu.ac.jp (Y.S.); ag173d11@dpc.agu.ac.jp (N.S.); ag183d03@dpc.agu.ac.jp (T.O.); ag193d11@dpc.agu.ac.jp (Y.S.); ag193d07@dpc.agu.ac.jp (S.K.); jun1row@dpc.agu.ac.jp (J.-i.H.); minita@dpc.agu.ac.jp (A.M.)

**Keywords:** interleukin-35, osteoimmunology, inflammatory bone destruction

## Abstract

Interleukin (IL)-35 is an immunosuppressive cytokine mainly produced by regulatory T cells. IL-35 mediates immunological functions by suppressing the inflammatory immune response. However, the role of IL-35 in bone-destructive diseases remains unclear, especially in terms of osteoclastogenesis. Therefore, the current study investigated the synergistic effect of IL-35 on osteoclastogenesis that is involved the pathogeneses of periodontitis and rheumatoid arthritis. Osteoclastic differentiation and osteoclastogenesis of RAW264 (RAW) cells induced by receptor activator of nuclear factor (NF)-κB ligand (RANKL) and IL-35 were evaluated by tartrate-resistant acid phosphate staining, hydroxyapatite resorption assays, and quantitative polymerase chain reaction. The effect of IL-35 on RANKL-stimulated signaling pathways was assessed by Western blot analysis. Costimulation of RAW cells by RANKL and IL-35 induced osteoclastogenesis significantly compared with stimulation by RANKL alone. Phosphorylations of extracellular signal-regulated kinase (ERK) and p38 mitogen-activated protein kinase tended to be increased by RANKL and IL-35 compared with RANKL or IL-35 alone. Additionally, the osteoclastogenesis induced by RANKL and IL-35 was suppressed by inhibition of ERK. In this study, IL-35 and RANKL induced osteoclastogenesis synergistically. Previous reports have shown that IL-35 suppresses the differentiation of osteoclasts. Therefore, IL-35 might play dual roles of destruction and protection in osteoclastogenesis.

## 1. Introduction

Bone-destructive diseases, including rheumatoid arthritis, osteoarthritis, osteoporosis, and periodontitis, are caused by inflammation or infection in diseased sites and lead to an imbalance of bone homoeostasis, resulting in severe bone destruction [1,2,3,4]. Bone homeostasis is affected by immune cells and cytokines through stimulating osteocytes, osteoblasts, and osteoclasts [5]. It is currently obvious that bone and the immune system are closely connected and their relationship is studied as osteoimmunology [6,7]. Osteoclasts and osteoblasts are mainly involved in bone resorption and the receptor activator of nuclear factor (NF)-κB ligand (RANKL)/RANK/osteoprotegerin (OPG) pathway plays a decisive role in the crosstalk between bone-destructive osteoclasts and osteogenic osteoblasts. Osteoclastogenesis is initiated by recognition of RANKL in osteoclast precursor cells, which is a critical regulator of osteoclastogenesis expressed on the osteoblast surface. Interleukin (IL)-17 secreted by T helper (TH) 17 cells has been widely recognized as a key modulator in autoimmune and infective diseases that involve local inflammation through the production of proinflammatory cytokines [8]. IL-17 and TH17 cells have been shown to promote osteoclastogenesis. IL-17 acts on osteoblasts by stimulating expression of RANKL, and TH17 cells secrete RANKL [9].

The IL-12 family is characterized by only consisting of heterodimeric cytokines including IL-12, IL-23, IL-27, and IL-35 [10,11]. IL-12 and IL-23 are chiefly proinflammatory cytokines with crucial roles in the development of TH1 and TH17 cells, respectively [12,13,14]. In contrast, IL-35, an immunosuppressive cytokine mainly produced by regulatory T cells (Tregs), mediates immunological functions by suppressing the inflammatory immune response [10].

IL-35 consists of the IL-12p35 subunit and IL-27β subunit Epstein–Barr virus-induced 3 (EBI3) [15,16]. It signals through a unique heterodimer of receptor chains IL-12Rβ2 and gp130 or homodimers of each chain [17]. IL-35 expression is affected by several immune and inflammatory states such as rheumatoid arthritis, inflammatory bowel disease, systemic sclerosis, asthma, chronic obstructive pulmonary disease, and periodontal disease [4,18,19,20,21,22]. Numerous studies related to the expression or function of IL-35 in bone-destructive disease have been reported recently. For example, IL-35 and IL-17 in gingival crevicular fluid (GCF) were significantly higher in patients with chronic periodontitis (CP) than in healthy subjects [4]. IL-35 is also elevated in CP gingiva compared with healthy tissue [23]. In contrast, a previous study showed that IL-35 injection attenuated collagen-induced arthritis (CIA) in mice with simultaneous suppression of IL-17 [24]. In addition, serum IL-35 and the number of Tregs were decreased considerably in patients with the moderate and high disease activity of rheumatoid arthritis [1].

However, the role of IL-35 in bone-destructive diseases remains to be elucidated, especially in terms of osteoclastogenesis. This study examined the function of IL-35 in osteoclastogenesis using monocytes only and investigated the pathological mechanism of IL-35 for the development of clinical therapies for bone-destructive diseases.

## 2. Results

### 2.1. IL-35 Synergistically Induces RANKL-Dependent Osteoclastogenesis of RAW264 (RAW) Cells

We stimulated RANKL-treated RAW cells with or without IL-35 to determine whether IL-35 affects RANKL-dependent osteoclastogenesis. RANKL treatment effectively induced differentiation of multinucleated osteoclasts, and interestingly, addition of 10, 50, or 100 ng/mL IL-35 synergistically and significantly induced the formation of tartrate-resistant acid phosphate (TRAP)-positive multinucleated cells (Figure 1). The multinucleated osteoclasts were observed to became larger in size and had increased numbers of nuclei after addition of IL-35 in a dose-dependent manner compared with RANKL alone (Figure 1A). However, 100 ng/mL IL-35 alone did not induce any differentiation of multinucleated osteoclasts.

### 2.2. IL-35 Synergistically Induces Activation of RANKL-Stimulated Osteoclasts

RAW cells were seeded on a Corning Osteo Assay Surface and stimulated by RANKL with or without IL-35 for 5 days to evaluate the efficacy of IL-35 to promote osteoclast activation. RANKL and IL-35 stimulated RAW cells to differentiate into activated osteoclasts and partially resorbed Osteo Assay Surfaces (Figure 2A). The total number of resorbed holes by osteoclasts treated with RANKL and IL-35 was significantly higher than that by osteoclasts treated with RANKL alone (Figure 2A). The addition of IL-35 increased the numbers of pits in a dose-dependent manner, which were significantly higher compared with RANKL alone (10 ng/mL: *p* < 0.05; 50 and 100 ng/mL: *p* < 0.01) (Figure 2B). However, the addition of 100 ng/mL IL-35 alone did not induce any resorbed holes.

### 2.3. IL-35 Synergistically Induces Regulation of Osteoclastic Markers in RANKL-Stimulated RAW Cells

RANKL-inductive specific genes are associated with osteoclast differentiation. Quantitative polymerase chain reaction (qPCR) analysis was used to determine the effect of IL-35 on expression of RANKL-inductive specific genes. RANKL significantly increased the mRNA expression of matrix metalloproteinase (MMP)-9, cathepsin K, TRAP, and nuclear factor of activated T cells (NFAT) c1 in RAW cells, and the addition of IL-35 induced expression all RANKL-inductive specific genes in a dose-dependent manner. Notably, the expression of genes modulated by RANKL together with 100 ng/mL IL-35 was significantly higher compared with that by RANKL alone (*p* < 0.01) (Figure 3). However, the addition of 100 ng/mL IL-35 alone did not induce any specific genes. In addition, chloride channel 7 (CLCN7) gene expression did not change with any stimuli in this study. These results indicate that IL-35 induces certain genes associated with osteoclast differentiation induced by RANKL.

### 2.4. Signaling Pathways Associated with the Acceleration of RANKL-Dependent Osteoclastogenesis by IL-35

We next examined the effect of IL-35 on RANKL-stimulated signaling pathways including extracellular signal-regulated kinase (ERK), c-jun N-terminal kinases (JNK), p38 mitogen-activated protein kinase (MAPK), and NF-κB by Western blot analysis. Stimulation by RANKL with or without IL-35 at 5 min significantly increased phosphorylations of ERK and p38 MAPK compared with no stimulation (control) (Figure 4). Stimulation by RANKL together with or without IL-35 at 10 min significantly increased phosphorylations of NF-κB compared with the control. The addition of RANKL and IL-35 tended to increase these phosphorylations compared with RANKL alone. Additionally, IL-35 alone tended to slightly induce phosphorylations of ERK, p38, and NF-κB compared with the control.

### 2.5. IL-35 May Enhance RANKL-Dependent Osteoclastogenesis via the ERK Signaling Pathway

RAW cells were pretreated in the presence or absence of a specific inhibitor for each kinase during osteoclastic differentiation mediated by RANKL and IL-35 to investigate which pathway was predominant for osteoclastogenesis induced by RANKL and IL-35. Pretreatment with each specific inhibitor, 10 μM SB203580 (p38 MAPK), PD98059 (ERK), and SP600125 (JNK), significantly inhibited the formation of RANKL-induced TRAP-positive multinuclear cells (Figure 5A). Pretreatment with PD98059 significantly inhibited RANKL and IL-35-induced TRAP-positive multinuclear cells, indicating that the ERK signaling pathway might be one of the elements of RANKL/IL-35-dependent osteoclastogenesis (Figure 5A,B).

## 3. Discussion

In this study, we demonstrated that IL-35 and RANKL synergistically induced osteoclastogenesis, including the pit-forming activity of osteoclasts, whereas IL-35 alone had no significant effect on osteoclastogenesis. In addition, the ERK signaling pathway may be a major element of RANKL/IL-35-mediated osteoclastogenesis.

It has been reported that specific genes, such as cathepsin K, MMP-9, and TRAP, are upregulated in response to RANKL [25,26,27]. Our results demonstrated that the expression of these osteoclast-specific genes was increased by RANKL, and IL-35 enhanced such gene expression similarly to other cytokines including IL-17 and IL-15 [28,29].

Previous reports have demonstrated that p38 MAPK plays a key role in RANKL-induced osteoclastogenesis [30,31]. Our results showed that IL-35-promoted osteoclastogenesis might be mediated by ERK in addition to p38 MAPK (Figure 5). The ERK pathway reacts to a broad range of extracellular stimuli and is a key mechanism by which extracellular signals are transmitted intracellularly [32]. Yokota et al. reported that ERK inhibitors PD98059 and U0126 significantly block differentiation of osteoclasts generated by addition of both TNFα and IL-6 [33]. In particular, the IL-6 signaling pathway downstream of JAK, corresponding to the MEK/ERK pathway, appears to play a critical role in the induction of c-Fos [34,35], which leads to activation of the osteoclast master regulatory transcriptional factor NF-ATc1 [36]. In this study, it remains unknown how important and how the IL-35-related ERK signaling pathway is involved in osteoclastogenesis. Our next study will elucidate the potential mechanism of ERK signaling pathways, which lead to osteoclastogenesis.

In contrast to our results indicating that IL-35 promoted RANKL-induced osteoclastogenesis, Jing et al. reported that IL-35 suppresses the differentiation of osteoclasts from mouse bone marrow derived-macrophage (BMMs) [23]. In addition, IL-35 inhibits TNFα-induced osteoclast differentiation and bone resorption of BMMs [37]. This discrepancy might be caused by the difference between each culture system used in the experiments. We have reported a similar phenomenon, namely that the effect on osteoclastogenesis by IL-15 is completely opposite depending on the culture system using a mouse monocyte cell line (IL-15 promotes RANKL-induced osteoclastogenesis) or BMMs (IL-15 inhibits PGE2-induced osteoclastogenesis) [28,38]. BMMs include many cell populations, mainly lymphocytes, which are affected by IL-35 in various manners. Thus, our results using a cell line showed differences from other reports using BMMs.

Moreover, Yago et al. reported that IL-35 inhibits RANKL-induced differentiation and activation of human osteoclasts derived from monocytes [39]. Blood leukocyte gene expression profiles induced by miscellaneous inflammatory events, including burns, endotoxemia, sepsis, and trauma, in mouse models and human subjects have indicated that gene expression fluctuations in the corresponding mouse models differed conspicuously from those in humans [40]. The current study showed that IL-35 promoted RANKL-induced osteoclastogenesis using mouse cell line RAW. Therefore, species differences may have contributed to this discrepancy in immune responses and altered transcriptional responses. Taken together, IL-35 might have dual effects in osteoclastogenesis. The current results suggest a novel effect of IL-35 in bone-destructive diseases such as CP and rheumatoid arthritis. Our previous report showed significantly higher IL-35 in GCF and gingival tissues from CP patients than in those from healthy participants [4]. Another study also demonstrated that IL-35 in GCF and serum of CP patients was significantly higher than that in the control group [41]. Non-surgical periodontal therapy correspondingly decreased the IL-35 level in GCF compared with untreated CP patients [42], suggesting an important role of IL-35 in chronic inflammation of CP. However, IL-35 and periodontal clinical indicators have been correlated negatively [43]. In terms of arthritis, IL-35 upregulates OPG and suppresses RANKL expression in rats with CIA and in cultured fibroblast-like synoviocytes [44]. IL-35 also inhibits IL-17 production in rats with CIA [24]. These results suggest that IL-35 plays a protective role against periodontal disease by maintaining immune system homeostasis and dampening the inflammatory response. In contrast, a report demonstrated that IL-35 gene transfer significantly increased the clinical scores of CIA [45]. Moreover, administration of IL-35 to Borrelia-vaccinated and -infected mice augmented the development of severe arthritis compared with untreated control mice [46]. These results suggest a proinflammatory effect of IL-35 in disease development. The current results demonstrate that IL-35 may also be involved in the proinflammatory tissue destruction of periodontitis and arthritis, and does not play a protective role. Therefore, IL-35 might play both destructive and protective roles and balance osteoclastogenesis by stimulating osteoclast precursor cells directly and suppressing IL-17 production.

In conclusion, this study demonstrates that IL-35 and RANKL synergistically induce osteoclastogenesis, and IL-35 might have a novel and important role in destructive bone diseases with inflammation such as periodontitis and rheumatoid arthritis. More detailed studies are required to assess the physiological functions of IL-35 in vitro and in vivo.

## 4. Materials and Methods

### 4.1. Cell Culture

Mouse monocyte cell line RAW was purchased from the RIKEN Cell Bank (Ibaraki, Japan) [47] and cultivated in α-minimum essential medium (Gibco, Yokohama, Japan) containing 10% (*v*/*v*) fetal bovine serum (MP Bio, Tokyo, Japan), 100 U/mL penicillin, and 100 μg/mL streptomycin at 37 °C with 5% CO_2_. The cells were seeded in 24-well culture plates at 1.3 × 10^4^ cells/well. RAW cells were cultured in medium with or without 50 ng/mL soluble recombinant RANKL (PeproTech, Rocky Hill, NJ, USA) and 10, 50, or 100 ng/mL soluble recombinant IL-35 (PeproTech) for 5 days with a medium change on day 3.

### 4.2. TRAP Staining

After 5 days of stimulation with or without RANKL (50 ng/mL) and IL-35 (10, 50, or 100 ng/mL), the cells were fixed with 3.7% formaldehyde for 5 min and then stained using an Acid Phosphatase, leukocyte (TRAP) Kit (Sigma-Aldrich, St. Louis, MO, USA) for 5 min at 37 °C. After washing with distilled water, the cells were viewed under a light microscope and TRAP-positive multinucleated cells with three or more nuclei were counted.

### 4.3. Hydroxyapatite Resorption Assay

A hydroxyapatite resorption assay was performed using the Corning Osteo Assay Surface (Corning Lifesciences, Corning, NY, USA) [48]. RAW cells were cultured on the 24-well Corning Osteo Assay Surface with or without RANKL (50 ng/mL) and IL-35 (10, 50, or 100 ng/mL) for 5 days. The wells were rinsed using 1.2% sodium hypochlorite and then the resorption area was visualized under a light microscope.

### 4.4. qPCR Analysis

RAW cells were incubated with or without RANKL (50 ng/mL) and IL-35 (10, 50, or 100 ng/mL) for 2 days. Total RNA was isolated from cultured cells using a NucleoSpin^®^ RNA (Macherey-Nagel Inc., Bethlehem, PA, USA), according to the manufacturer’s instructions. cDNA was synthesized from the total RNA by the action of ReverTra Ace^®^ (Toyobo Co. Ltd., Osaka, Japan). TaqMan gene expression assays were used to measure MMP-9 (Mmp9:Mm00442991-m1), cathepsin K (Ctsk:Mm00484039-m1), TRAP (Acp5:Mm00437135-m1), NFATc1 (Nfatc1:Mm00479445-m1), and CLCN7 (Clcn7:Mm00442400-m1) expression using TaqMan Universal PCR master mix (Thermo Fisher Scientific, Wilmington, DE, USA). 18S rRNA (Hs99999901-s1) was used for normalization of target gene expression levels. Reactions were performed with initial denaturation at 95 °C for 10 min, followed by 40 cycles of 95 °C for 15 s and 60 °C for 1 min using the ABI Prism 7000 sequence detection system (Thermo Fisher Scientific). Quantification of mRNA for the target cytokine was conducted using the cycle threshold (CT) values of the target gene and 18S rRNA. The fold change in mRNA expression levels was determined as follows: 2^−ΔΔCt^, where ΔΔCT = [(CT of target − CT of 18S rRNA (treated group)) − (CT of target − CT of 18S rRNA (control group))].

### 4.5. Western Blot Analysis

RAW cells were lysed in CelLyticM lysis buffer (Sigma-Aldrich) with protease and phosphate inhibitor cocktails (Nacalai Tesque, Kyoto, Japan). Proteins were separated by sodium dodecyl sulfate-polyacrylamide gel electrophoresis (Bio-Rad Laboratories Hercules, CA, USA), transferred onto polyvinylidene fluoride membranes (Bio-Rad Laboratories), and probed with an anti-mouse p44/42 MAPK (Erk1/2) rabbit mAb, anti-mouse phospho-p44/42 MAPK (Erk1/2) (Thr202/Tyr204) (D13.14.4E) XP rabbit mAb, anti-mouse SAPK/JNK rabbit mAb, anti-mouse phospho-SAPK/JNK (Thr183/Tyr185) (81E11) rabbit mAb, anti-mouse p38 MAPK rabbit monoclonal antibody (mAb), anti-mouse phospho-p38 MAPK (Thr180/Tyr182) (D3F9) XP rabbit mAb, anti-mouse-NF-κB p65 (D14E12) XP rabbit mAb, anti-mouse phospho-NF-κB p65 (Ser536) (93H1) rabbit mAb, or anti-mouse β-actin (8H10D10) mouse mAb (all purchased from Cell Signaling Technologies, Danvers, MA, USA). Horseradish peroxidase-labeled anti-rabbit IgG, HRP-labeled anti-mouse IgG (Cell Signaling Technologies), and goat anti-mouse IgG-HRP (Santa Cruz Biotechnology) were used as secondary antibodies. ImmunoStar LD (WAKO, Tokyo, Japan) and the RAS-3000 System were used to visualize the proteins.

### 4.6. Effects of MAPK and NF-κB Inhibitors on RANKL/IL-35-Dependent Osteoclastogenesis

Cells were pretreated with 10 μM MAPK and NF-κB inhibitors MG-132 (proteasome inhibitor), SB203580 (p38 MAPK specific), PD98059 (ERK specific), or SP600125 (JNK specific) for 1 h. After pretreatment, the cells were incubated with 50 ng/mL RANKL and 100 ng/mL IL-35 for 5 days and then subjected TRAP staining as described above.

### 4.7. Statistical Analysis

SPSS 12.0 software (SPSS Inc., Chicago, IL, USA) was used to perform statistical analyses. Data are expressed as the mean ± SD and were analyzed using one-way and two-way analysis of variance (ANOVA). A *p*-value of <0.05 was considered as significant.

## Figures and Tables

**Figure 1 ijms-21-02069-f001:**
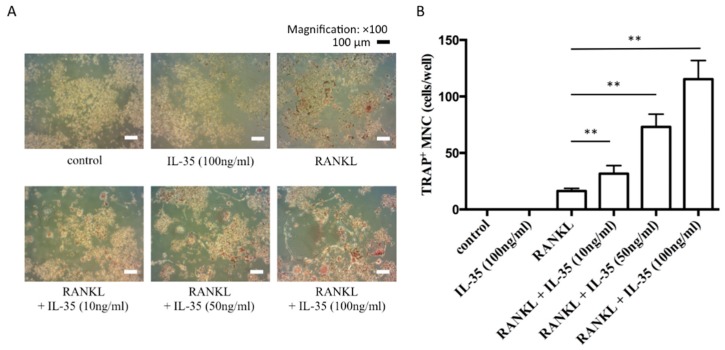
IL-35 synergistically induces receptor activator of nuclear factor-κB ligand (RANKL)-dependent osteoclast formation. (**A**) RAW264 (RAW) cells stimulated with 50 ng/mL RANKL with or without 10, 50, or 100 ng/mL interleukin (IL)-35 or 100 ng/mL IL-35 alone for 5 days. Bar, 100 µm. (**B**) Tartrate-resistant acid phosphate (TRAP)-positive multinucleated cells were counted as osteoclasts. Differences among groups were analyzed by two-way ANOVA. Data represent the mean ± SD (*n* = 3) ** *p* < 0.01 vs. RANKL alone.

**Figure 2 ijms-21-02069-f002:**
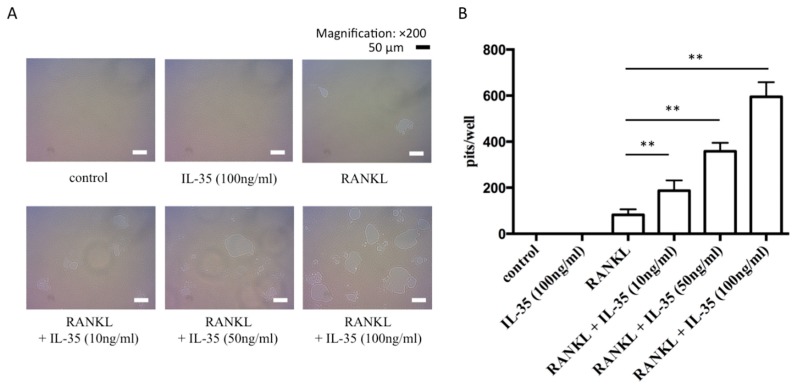
IL-35 synergistically induces RANKL-dependent osteoclast activation. (**A**) RAW cells stimulated with 50 ng/mL RANKL with or without 10, 50, or 100 ng/mL IL-35 or 100 ng/mL IL-35 alone for 5 days on the Corning Osteo Assay Surface. Bar, 50 µm. (**B**) Formed pits were counted to indicate osteoclast activation. Differences among groups were analyzed by two-way ANOVA. Data represent the mean ± SD (*n* = 3). ** *p* < 0.01 vs. RANKL alone.

**Figure 3 ijms-21-02069-f003:**
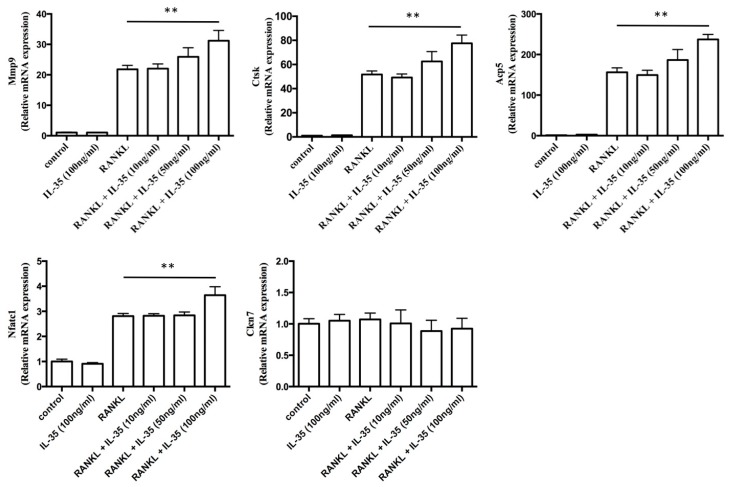
IL-35 synergistically induces expression of RANKL-dependent osteoclast-specific markers. matrix metalloproteinase (MMP)-9 (Mmp9), cathepsin K (Ctsk), TRAP (Acp5), nuclear factor of activated T cells (NFAT) c1 (Nfatc1), and chloride channel 7 (CLCN7) (Clcn7) mRNA expression was investigated in RAW cells stimulated with 50 ng/mL RANKL together with or without 10, 50, or 100 ng/mL IL-35 or 100 ng/mL IL-35 alone for 2 days by qPCR. Values are expressed as fold changes. Differences among groups were analyzed by two-way ANOVA. Data represent the mean ± SD (*n* = 3). ** *p* < 0.01 vs. RANKL alone.

**Figure 4 ijms-21-02069-f004:**
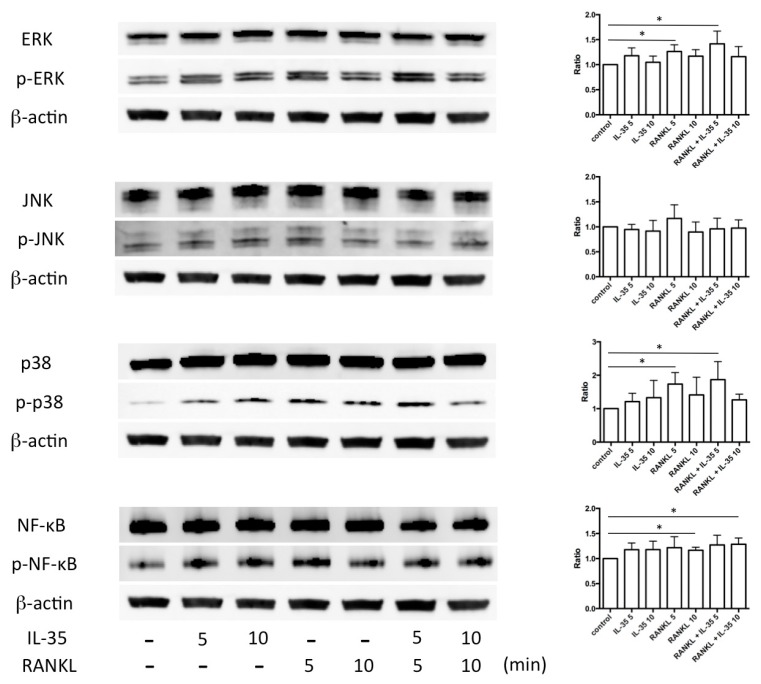
RANKL and IL-35 stimulate several signaling pathways. Left: extracellular signal-regulated kinase (ERK), p-ERK, c-jun N-terminal kinases (JNK), p-JNK, p38, p-p38, nuclear factor (NF)-κB, and p-NF-κB were measured in RAW cells stimulated with 50 ng/mL RANKL together with or without 100 ng/mL IL-35 for the indicated times by Western blot analysis. Right: The ratio of the phosphorylated forms to the unphosphorylated forms was quantified in three blots using ImageJ. Differences among groups were analyzed by two-way ANOVA. Data represent the mean ± SD (*n* = 3). * *p* < 0.05.

**Figure 5 ijms-21-02069-f005:**
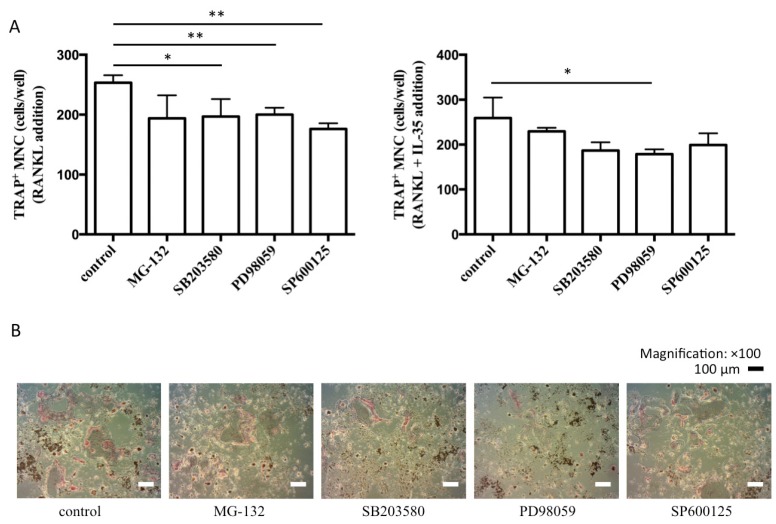
ERK signaling pathway is associated with IL-35-dependent osteoclastogenesis. (**A**) RAW cells were pretreated with each specific inhibitor for 1 h and then stimulated with 50 ng/mL RANKL together with or without 100 ng/mL IL-35 for 5 days. TRAP-positive multinucleated cells were counted as osteoclasts. Differences among groups were analyzed by one-way ANOVA. Data represent the mean ± SD (*n* = 3). * *p* < 0.05, ** *p* < 0.01 vs. control. (**B**) RAW cells were pretreated with each specific inhibitor and then stimulated with 50 ng/mL RANKL and 100 ng/mL IL-35. After 5 days, the cells were fixed and stained for TRAP.

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
