# Peer review of "IL-35 and RANKL Synergistically Induce Osteoclastogenesis in RAW264 Mouse Monocytic Cells"

_ijms, 2020, doi:10.3390/ijms21062069_

Round 1
Reviewer 1 Report
The authors have addressed all points raised in the previous review and have greatly improved the manuscript. I commend the authors for performing the additional experiments and providing additional data. However, one major concern still remains. The data presented does not support the conclusion that the synergistic effect of IL-35 and RANKL is mediated via ERK signaling. Indeed, while ERK signaling is modulated, this effect is very similar to RANKL-only treatment. Therefore, I would recommend that the authors soften their conclusion and discuss ERK-signaling as a potential mechanism that requires further investigation.
Author Response
Point-by-point responses to reviewer comments
Responses to Reviewer 1:
The authors have addressed all points raised in the previous review and have greatly improved the manuscript. I commend the authors for performing the additional experiments and providing additional data. However, one major concern still remains. The data presented does not support the conclusion that the synergistic effect of IL-35 and RANKL is mediated via ERK signaling. Indeed, while ERK signaling is modulated, this effect is very similar to RANKL-only treatment. Therefore, I would recommend that the authors soften their conclusion and discuss ERK-signaling as a potential mechanism that requires further investigation.
Response: We thank the reviewer for evaluating the effort involved in this revision, as well as any further valuable suggestion. In the revised manuscript, we have revised the descriptions of the involvement and importance of ERK signaling in Results and Discussion section (page 6, line 168, page 7 line 189, page 8, lines 197-199) and also delated the description in conclusion like below.
“In conclusion, this study demonstrates that IL-35 and RANKL synergistically induce osteoclastogenesis mainly through the ERK signaling pathway, and IL-35 might have a novel and important role in destructive bone diseases with inflammation such as periodontitis and rheumatoid arthritis. More detailed studies are required to assess the physiological functions of IL-35 in vitro and in vivo.”
Reviewer 2 Report
Congratulations to the authors for this interesting study.
Attention to the references, correct typos:
revise
reference 3
De Martinis, M.; Sirufo, M. M.; Ginaldi, L., Osteoporosis: Current and emerging therapies targeted to immunological checkpoints. Curr Med Chem 2019, 26, 1-16. doi: 10.2174/0929867326666190730113123.
and reference 6
Ciccarelli, F.; De Martinis, M.; Ginaldi, L., Glucocorticoids in patients with rheumatic diseases: friends or enemies of bone? Curr Med Chem 2015, 22, (5), 596-603.
Author Response
Point-by-point responses to reviewer comments
Responses to Reviewer-2:
Congratulations to the authors for this interesting study.
Attention to the references, correct typos:
revise
reference 3
De Martinis, M.; Sirufo, M. M.; Ginaldi, L., Osteoporosis: Current and emerging therapies targeted to immunological checkpoints. Curr Med Chem 2019, 26, 1-16. doi: 10.2174/0929867326666190730113123.
and reference 6
Ciccarelli, F.; De Martinis, M.; Ginaldi, L., Glucocorticoids in patients with rheumatic diseases: friends or enemies of bone? Curr Med Chem 2015, 22, (5), 596-603.
Response: We thank the reviewer for considering our study as interesting one. We apologize for this careless mistake, which has been corrected in the revised manuscript (page 11, lines 337-338, and page 12, lines 345).
Round 2
Reviewer 1 Report
All concerns have been addressed.
This manuscript is a resubmission of an earlier submission. The following is a list of the peer review reports and author responses from that submission.
Round 1
Reviewer 1 Report
In this manuscript Kamiya et al. investigate the effect of IL-35 in combination with RANKL on the osteoclastogenesis of RAW264 cells. The authors and other investigators have previously identified IL-35 to be involved in bone-destructive diseases, thus the aim of this study was to investigate a direct effect of IL-35 on osteoclast activity. The paper is very well written and the methods are in principal suitable to address the research question. The study is potentially interesting for the scientific community, however, due to the concerns listed below, I cannot recommend this manuscript for publication in its current form.
Major concerns:
The authors claim a synergistic effect of IL-35 and RANKL on the osteoclastogenesis of RAW264 cells. This cannot be concluded since no quantified data of the effect of IL-35 alone is presented. Thus the effect might be just additive. The authors need to provide data of only IL-35 treatment for all datasets.
Figure 2 is missing.
Gene expression data for additional osteoclast marker genes such as Nfatc1, Ap6i or Clcn7 should be provided.
The Western blots need to be repeated at least three times and quantified, to provide reliable data that can be statistically analyzed since the differences are only minor.
The data in Figure 5 is very inconclusive and even cast doubt on the previous results, as the combinatory effect of Rankl and IL-35 on the number of Trap+ MNC/well is no longer there. For both Rankl+IL-35 and Rankl alone this value is now around 250 instead of approx. 75 vs. 15 as shown before. Furthermore, the effects of the inhibitors in Fig. 5A are very similar between Rankl alone and Rankl+IL-35. The different significances observed appear to be caused be differences in the variation of the control group and not in the actually treated groups. The sample size needs to be increased and Rankl and Rankl+IL35 groups need to be directly compared to draw appropriate conclusions regarding the underlying signaling pathway. As it stands now, the role of the ERK signaling pathway is highly doubtful.
Since the data are only limited to the RAW264 cell line, a repetition of key experiments with e.g. bone marrow cells would greatly improve the impact and translatability of the results and conclusions.
Minor concerns:
Two-way ANOVA is a more appropriate statistical method to discern the effect of Rankl and IL35 individually.
What is the effect of these treatments on cell morphology? (size, number of nuclei)
Please avoid large empty areas in the figures.
p.5 of 12, line 142: “an” should be “a”
Reviewer 2 Report
The study is interesting and well thought-out.
IL-35 is a relatively new cytokine that emerges as an important immunomodulator. IL-35 appears to be anti-inflammatory and to protect from autoimmune inflammatory diseases, upregulates osteoprotegerin and suppresses RANKL, thus inhibiting bone resorption.
However I would like the authors to:
In the text the paragraph materials and methods is skipped at the end of the manuscript: revise
Correct typos and grammatical errors.